# Testing the feasibility and utility of an executive function battery for use with primary school-aged students in Malawi

**Michael T. Willoughby**[1]◉*, **Maclean Vokhiwa**[2,3]◉, **Amanda C. Wylie**[1], **Richard Reithinger**[4], **Lauren M. Cohee**[5]

**1** Education Practice Area, RTI International, United States of America, **2** Department of Pediatrics and Child Health, Kamuzu University of Health Sciences, Blantyre, Malawi, **3** Malaria Alert Centre, Kamuzu University of Health Sciences, Blantyre, Malawi, **4** International Development Group, RTI International, United States of America, **5** Department of Clinical Sciences, Liverpool School of Tropical Medicine, Liverpool, United Kingdom

◉ These authors contributed equally to this work.
* mwilloughby@rti.org

## Abstract

There is increasing evidence that malaria impacts student educational outcomes, in part through impairments in cognitive function. Currently, there is no consensus with regards to standardized tools or approaches to assess the extent and magnitude of this association. We conducted a pilot study to assess the feasibility and utility of a well-established tablet-based battery of executive function (EF) tasks for primary school-aged children in Malawi. **We** collected data from 197 students in grades 1–4 in a rural primary school in Blantyre District, Malawi. The assessment battery ("EF Touch"), which consisted of seven EF tasks that measure inhibitory control, working memory, and cognitive flexibility, was administered using open-source, standardized tablet-based software (RTI International's Tangerine). Assessments were conducted in Chichewa, and task performance was analyzed for accessibility and challenge across different grade levels. High completion rates were observed for all tasks, and most students completed the entire battery within one hour. Task performance varied by grade, with older students generally performing better. Two tasks had poor performance and ceiling effects and were omitted from composite scores. A composite of EF task performance was normally distributed and increased with grade level. The study demonstrates the feasibility of using a common battery of EF tablet-based assessments with students in grades 1–4 in Malawi. Given the high burden of malaria in this region and its potential impact on cognitive development, these results help to establish the feasibility and utility of direct EF assessments in future studies that focus on the impact of malaria infection on cognitive and educational outcomes.

**Data availability statement:** All relevant data are within the paper and its Supporting Information files.

**Funding:** We did not receive any external funding for this work. RTI International (which is the trade name for the Research Triangle Institute) provided internal institutional research and development funding to R.R. to conduct this work. The award of internal funds did not include any requirements or advice related to study design, data collection and analysis, the decision to publish, or the preparation of this manuscript.

**Competing interests:** The authors have declared that no competing interests exist.

## Introduction

Despite widespread implementation of malaria prevention and control interventions over the last 15–20 years, the burden of malaria remains high, particularly in areas of sub-Saharan Africa [1]. Early childhood (i.e., birth to 5 years of age) is a period of high risk of severe malaria and malaria-related mortality. Large investments in interventions that specifically target young children have resulted in substantial reductions in malaria prevalence and incidence in this age group [1]. Comparatively less attention has been devoted to the impact of malaria in school-aged children, even though the prevalence of malaria infection peaks in this age group in many endemic countries—consequently, they serve as a main reservoir of malaria infection that perpetuates community transmission [2–5].

New efforts are underway to increase malaria prevention and control efforts among school-aged children. In addition to addressing disease burden, there is growing interest in considering whether these health-focused efforts have secondary benefits on children's educational outcomes [6,7]. Understanding the impact of health interventions on educational outcomes highlights the potential for multisectoral collaborations, including the potential for these interventions to contribute to achieving Sustainable Development Goals 4.1.1 and 4.2.1 (Quality Education and Early Childhood Development).

Executive function (EF) skills are foundational cognitive processes that facilitate goal-directed behavior and support academic learning by allowing one to think flexibly (cognitive flexibility or attention shifting), hold information in mind and update it (working memory), and overcome highly learned or prepotent responses (inhibitory control) [8]. EF skills develop rapidly in early and middle childhood and are malleable. For example, exposure to poverty-related stressors (e.g., food, housing and income uncertainty), xenobiotics, and pathogens influence the structure and function of neural circuits and stress physiology systems that support EF skill development [9,10]. Neuroplasticity in the developing brain makes the circuits that support EF skills particularly vulnerable to social and biological stressors [11]. Multiple mechanistic pathways connect malaria infection specifically to impaired neurologic functioning; these include malarial anemia, parasite sequestration, endothelial dysfunction, and cerebral hypoxia with cerebral and uncomplicated malaria, vascular leakage, and neuroinflammation [12,13]. The extent of malaria-induced cognitive impairment varies as a function of malaria severity, chronicity, co-occurring risk factors (e.g., genetic disposition, malnutrition, socioeconomic status), cumulative number of infections, treatment, and age of (re)infection [12,14].

A recent systematic review and meta-analysis documented that severe malaria in early or middle childhood was associated with pronounced impairments in attention (standardized mean difference [SMD] = -.68), memory (SMD = -.52), and language (SMD) = -.36) relative to unaffected children [15]. Many studies in this review relied on general measures of cognition (e.g., Kaufman Assessment Battery for Children; Malawi Development Assessment Tool). Although a few studies included indicators of sustained attention and information processing, which are conceptually similar to EF skills, no study included a battery of tasks that measured the foundational domains of EF skills, including inhibitory control, working memory, and cognitive flexibility [16].

We have been iteratively developing a computerized battery of performance-based EF tasks (i.e., EF Touch) for over a decade. EF Touch presents children with a series of game-like tasks, each of which uses an established neuropsychological paradigm to measure a specific EF skill (e.g., go/no paradigm for inhibitory control; self-ordered paradigm for working memory). The EF Touch battery was initially delivered via computer (requiring a laptop or desktop that ran the Microsoft Windows operating system and a secondary touch-screen monitor). Support for the validity and psychometric properties of that version of the battery has been firmly established with preschool-aged children in the United States [17,18]. More recently, we adapted EF Touch for use in international settings using RTI International's Tangerine open-source software, which runs on tablets that use an Android operating system (i.e., Tangerine EF Touch). Evidence for the feasibility and predictive validity of the Tangerine EF Touch was established with preschool-aged children in Kenya [19,20]. Our work is part of a larger trend in the field of using tablet-based cognitive assessments with children in low-resourced settings [21].

Given the potential for malaria infection, even in the absence of severe disease, to interfere with normative EF skill development, we conducted a pilot study to test the feasibility and utility of an age-expanded version of Tangerine EF Touch for use with school-aged students. Specifically, we focused on students in grade levels 1–4, a group targeted for school-aged malaria interventions. Participating students in grades 1–4 in our study ranged from 5-13-years-old, which highlights the need to use tasks that accommodate a wide range of cognitive ability. Preprimary education is not compulsory in Malawi and approximately one-third of children participate in some type of early childhood education program [22]. Hence, for many students in grade 1, this was their first year of formal education. Moreover, given the high rate of poverty (70% of the population of Malawi lives below the international poverty line of $2.15/day) [23] and limited access to technology, we elected to modify Tangerine EF Touch tasks that had been used with preprimary-aged children (3–6-years). Three sets of task adaptations were made to accommodate the sampling of students in higher grades. First, timed tasks were accelerated to increase processing speed and inhibitory control demands. Second, tasks that involved multiple rules had those rules presented and practiced simultaneously to increase working memory demands. Third, we incorporated a tablet-based working memory task that was successfully used in previous studies of school-aged children in Malawi [24].

## Methods

### Participants

We collected data over a 3-week period in a single primary school in rural Blantyre District, Malawi, which was selected because of its proximity to the Kamuzu University of Health Sciences and community receptivity to prior malaria-related research. Written ethics approval was provided by Kamuzu University of Health Science's Research Ethics Committee (P.04/24–0737). The study coordinator engaged with local school leadership, who distributed information about the study to parents of students in grades 1–4. Study recruitment and data collection occurred from 05/07/2024–31/07/2024. For all children enrolled in the study, we obtained written informed consent from caregivers and verbal assent from students ≥10 years old. We completed pilot assessments with 17 students as a part of staff training and certification and subsequently collected data from 197 students during the main study. Our central question was whether a common set of EF tasks were sufficiently accessible for students in grade 1, while being sufficiently challenging for students in grade 4. As such, we employed a stratified sampling plan that over-sampled children in grades 1 and 4 (i.e., we recruited n = 69, 26, 31, and 71 students in grades 1–4, respectively). Participating students were on average 9.0 years old (SD = 2.2 years; Range = 5–13 years) and evenly distributed with respect to sex (48% female). All assessments were conducted in Chichewa. We provided reimbursements equivalent to USD $10 (MWK 17,000.00) to participants for transport, time, and refreshments.

### Measures

The EF assessment consisted of a battery of nine tasks ('games'). The first two tasks (training, bubbles) were intended to orient students to the structure of the assessment (i.e., assessors read standardized task instructions; students respond

PLOS Global Public Health

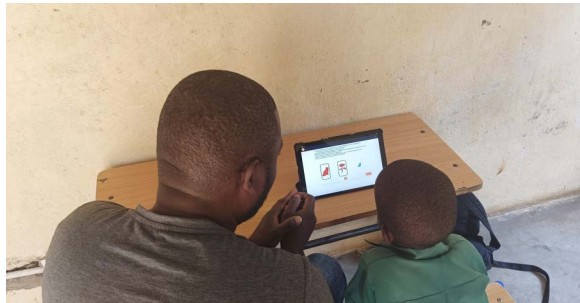

**Fig 1. Typical Setup for Tablet-Based EF Assessment. Depiction of an assessor-supported, individual child assessment.**

by touching the tablet; see Fig 1). The remaining seven tasks measured subdomains of EF using existing paradigms from the neuropsychological literature. All EF tasks shared a similar structure. Assessors described the task using standardized instructions and demonstrated correct answers. Students were then administered training items. If they passed training items (determined by program logic, not assessors), they proceeded to test items. Tasks were discontinued if students failed training items twice. All tasks that measure inhibitory control were speeded (i.e., each test item appears for up to 2500 milliseconds, and students are instructed to make responses as fast as they can). EF tasks were administered in a counter-balanced order (i.e., different task order for each assessor). Table 1 indicates the specific subdomains of EF (i.e., inhibitory control, working memory, cognitive flexibility) domains that were measured by each task. Full text descriptions of tasks are provided in S1 Table.

## Procedures

Task development and assessor training proceeded in four steps. First, we selected seven tasks that we had used in recent studies and that had previously been translated into Chichewa. With the exception of the Short Memory task (24), all tasks had been used in our earlier work. All EF tasks were administered using Tangerine software (https://www.tangerinecentral.org/). The software is an open-source platform that facilitates tablet-based data collection (school-based assessments occur offline, with files saved locally on the tablet and uploaded to a project server when staff

**Table 1. Descriptive statistics of student performance on six EF tasks.**

|  | Inhibitory Control | | Cognitive Flexibility | | Working Memory | | EF Composite |
|---|---|---|---|---|---|---|---|
|  | Silly Sounds | H&F | Something's Same | H&F | Pick Picture | Short Memory |  |
| **N** | 193 | 133 | 187 | 133 | 193 | 193 | 197 |
| **Mean (SD)** | 0.64 (0.25) | 0.82 (0.24) | 0.83 (0.11) | 0.71 (0.22) | 0.80 (0.09) | 0.23 (0.12) | 0.66 (0.13) |
| **Min, Max** | 0.13, 1.00 | 0.00, 1.00 | 0.43, 1.00 | 0.14, 1.00 | 0.50, 1.00 | 0.00, 0.54 | 0.29, 0.95 |
| **Correlations** |  |  |  |  |  |  |  |
| IC: Silly Sounds | -- | 0.35* | 0.31* | 0.32* | 0.32* | 0.46* | 0.78* |
| IC: H&F |  | -- | 0.22 | 0.52* | 0.26* | 0.28* | 0.73* |
| CF: Something's Same |  |  | -- | 0.15 | 0.28* | 0.37* | 0.55* |
| CF: H&F |  |  |  | -- | 0.28* | 0.41* | 0.74* |
| WM: Pick the Picture |  |  |  |  | -- | 0.39* | 0.52* |
| WM: Short Memory |  |  |  |  |  | -- | 0.66* |

Note. H&F = Hearts and Flowers; IC = Inhibitory Control; CF = Cognitive Flexibility; WM = Working Memory. *$p < 0.01$

returned to the research center each day). Staff with in-country experience vetted the task instructions (language) and images for student familiarity. Second, a train-the-trainer model was employed. A US-based staff (MW) trained the study coordinator (MV) on how to administer and upload tasks; the coordinator subsequently trained and certified five assessors, all of whom had previous in-country experience with child assessments and task adaptation. Third, assessors completed 2–3 pilot cases and engaged in group debriefing with MW and MV (consensus decisions were reached about how to handle unexpected events or child responses). Pilot data were reviewed for accuracy. Fourth, assessors engaged in data collection for approximately 15 days at the end of the school year. All assessments were individually administered in the school setting and during typical school hours. Students were given opportunities to take breaks between tasks; assessors could also suggest short breaks if a student appeared tired or bored. At the completion of each task with each child, assessors completed a qualitative rating of task quality (i.e., low, OK, high), which reflected their impression of student comprehension of and engagement with the task, as well as the testing environment (e.g., distractions). At the end of each day, assessors completed a form that indicated which students were tested and their overall impressions of data collection. We reviewed daily notes for recurring themes. Deidentified data are available in S1 Data.

### Inclusivity in global research

Additional information regarding the ethical, cultural, and scientific considerations specific to inclusivity in global research is included in the Supporting Information.

## Results

### Overview

The rates of task completion were uniformly high (from 94% [Hearts and Flowers] to 99% [Short Memory]). Nearly all (94%) students completed at least six of the seven EF tasks (59% completed seven tasks; 35% completed six tasks; 5% completed five tasks; 1% completed only four tasks). Task quality ratings were also uniformly high (between 95% [Spatial Conflict Arrows] and 99% [Something's the Same] of tasks were rated as "OK" or "high" quality). The entire assessment was typically completed in less than one hour ($Mn = 47$ minutes; Range $= 32$–$64$ minutes).

### EF task performance

Based on preliminary analyses, we omitted the Spatial Conflict Arrows task. Students performed poorly on the congruent items, which do not require inhibitory control; this suggested that many students were unfamiliar with an arrow as a symbol that denotes direction. We also omitted the Animal Go/No Go task due to excessive ceiling effects (57% students completed 100% of no-go items correctly). This resulted in two scores each for the constructs of inhibitory control, working memory, and cognitive flexibility. Task accuracy descriptive statistics are presented in **Table 1** and score distributions are presented in **Fig 2**. We observed excellent variability in accuracy scores, with ranges between 50% [Pick the Picture] and 100% [Hearts and Flowers inhibitory control]). We observed low overall rates of floor and ceiling effects, defined as accuracy scores ≤5% or ≥95%, respectively. For five of the six tasks, fewer than 5% of students earned a score at or below the floor threshold and fewer than 15% of students earned a score at or above the ceiling threshold. For the Hearts and Flowers inhibitory control score, 44% of students scored at or above the ceiling threshold. However, because this is a timed task, we combined item accuracy and reaction time metrics to generate a score that did not suffer from ceiling effects (see S1 Fig,S2 Fig).

**Fig 2** also demonstrated that task performance improved as a function of student grade. Task accuracy scores did not differ by gender (not presented).

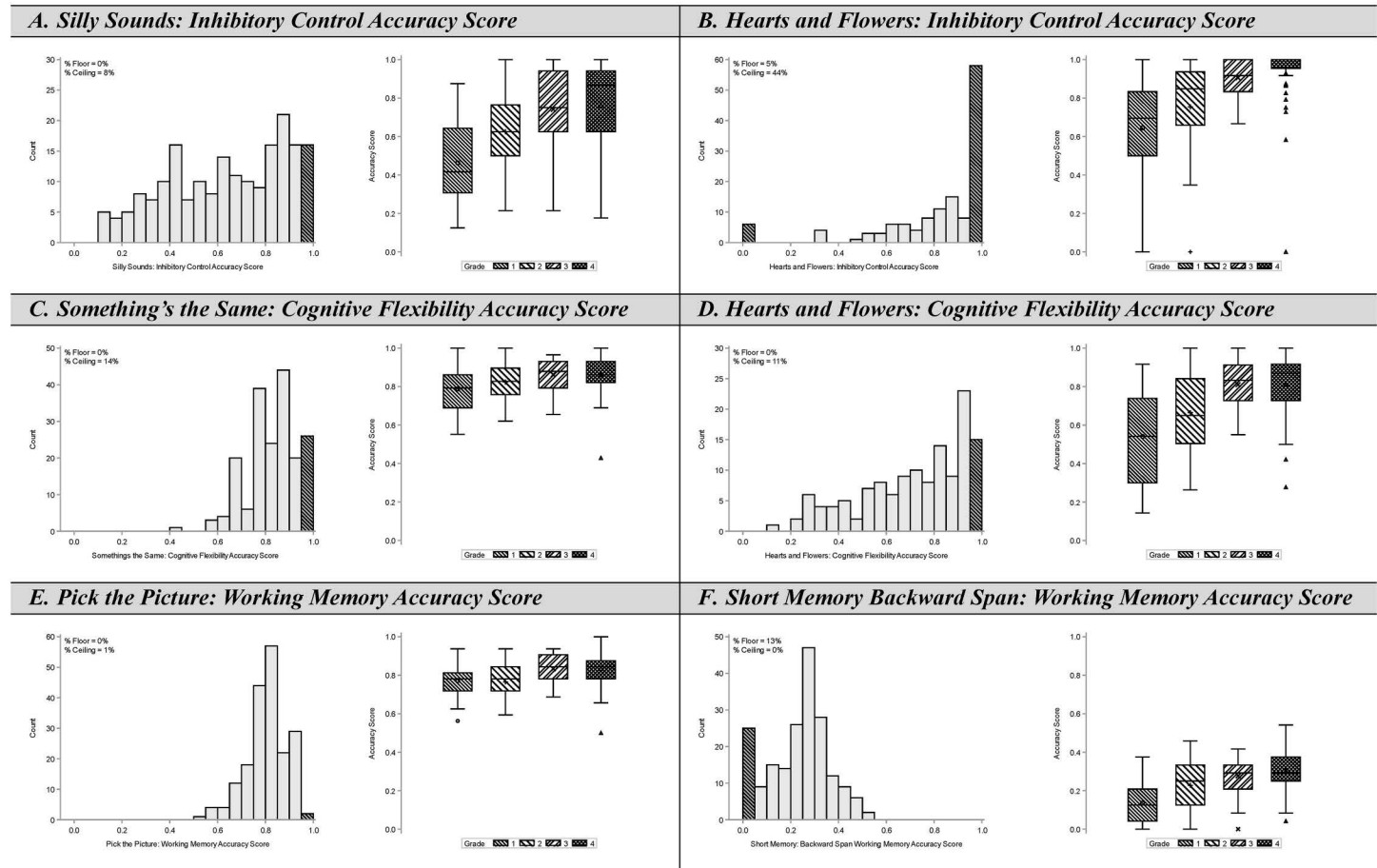

**Fig 2. Score Distributions for Individual EF Tasks in Total Sample and by Grade-level.** For each task, left panels depict histograms for the total sample, including the proportion of students with floor or ceiling effects, while right panels are box and whisker plots by grade level.

## EF composite

Task accuracy scores tended to be moderately correlated ($rs = 0.15-0.52$; see **Table 1**). These moderate correlations suggest that EF tasks are capturing complementary cognitive processes. Three EF construct scores (inhibitory control, cognitive flexibility, working memory) were generated by taking a mean of the representative task scores (two each). We then combined these scores into a single composite score by taking a simple mean. For most of the sample ($N = 192$), the EF score reflects an equal contribution of inhibitory control, cognitive flexibility, and working memory. For a smaller proportion of the sample ($N = 5$ or 3%), the EF score reflects an equal contribution of inhibitory control and working memory (the cognitive flexibility score could not be computed). The EF composite score ($Mn = 0.66$, $SD = 0.13$) ranged from 0.29 to 0.95. As shown in **Fig 3**, this score was normally distributed (with a larger grouping of scores around 0.75-0.80), increased with grade as expected, and only 1 student (in the 4th grade) obtained an overall score that reached the ceiling threshold of ≥95%.

## Challenges in EF assessment

A thematic network analysis of assessor impressions identified two sets of assessment challenges (**Fig 4**). The first set involved technical and device challenges (e.g., lack of familiarity with how to touch tablet screen; instances of apparent

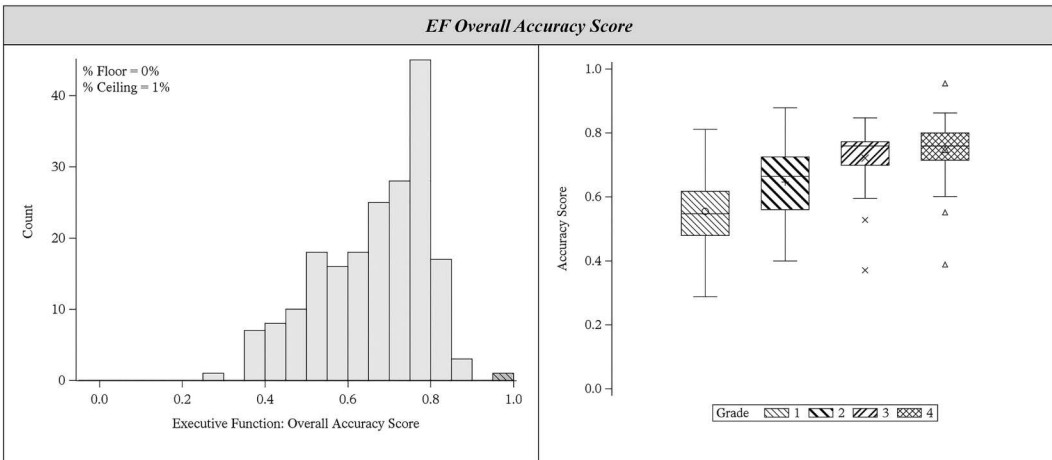

**Fig 3. Score Distribution for EF Composite in Total Sample and by Grade-level.** Left panel is the histogram for the EF composite score, including the proportion of students with floor or ceiling effects. Right panel is a box and whisker plot for the EF composite score by grade level.

tablet non-responsiveness or malfunction). The second set involved student behavioral challenges (e.g., poor concentration, fatigue, general developmental concerns).

## Discussion

The prospect that infections such as malaria undermine the development of EF skills, which support academic learning, highlights the need for performance-based measures of EF for use in future studies and clinical trials. Here, we demonstrated the feasibility and initial promise of direct tablet-based assessment of EF skills with primary school-aged students in Malawi. Most students completed the entire battery of EF tasks, and assessments took less than one hour. Students performed well on all tasks, with expected age-related differences in performance, and minimal evidence of floor or ceiling effects. Assessors viewed most assessments as having acceptable quality. A consistent theme to emerge from daily assessor notes was that a minority of students experienced difficulties in having their touches recognized by the tablet. Further adaptations addressing these challenges may enhance assessment accuracy and engagement.

Based on these data, we recommend four changes to this battery of EF tasks for use in studies that include school children in grades 1–4. First, we recommend omitting the Arrows and Animal Go/No-Go tasks for future use in this age group. Poor performance on the congruent items of the Arrows tasks suggested that a subset of students was unfamiliar with arrows as a symbol (at least the symbol used here) that denotes direction. The Hearts and Flowers tasks is conceptually similar and provided higher quality information. The Animal Go/No-Go task was limited by ceiling effects. Unlike other speeded tasks, item level reaction time information was uninformative of ability (i.e., we only know how quickly a child makes an incorrect response for no-go items). By administering multiple EF tasks designed to measure the same EF construct, we can modify the EF battery while ensuring that construct measurement does not depend solely on a single task score. Second, assessor notes indicated that a subset of (especially younger) students had difficulty having their touches recognized by the tablet. There are implicit fine motor skills that are required to touch tablets (e.g., related to the pressure and duration of touch) that may depend on previous interactions with smart phones and tablets. To the extent that some students have not had these experiences, this may contribute measurement error to their task performance. We recommend developing an extended training task that provides students with direct instruction and opportunities to practice touches prior to engaging in EF tasks. Third, a few of the inhibitory control tasks (especially Hearts and Flowers) exhibited more ceiling effects than is desirable. Although we can make combined use of item-level accuracy and reaction

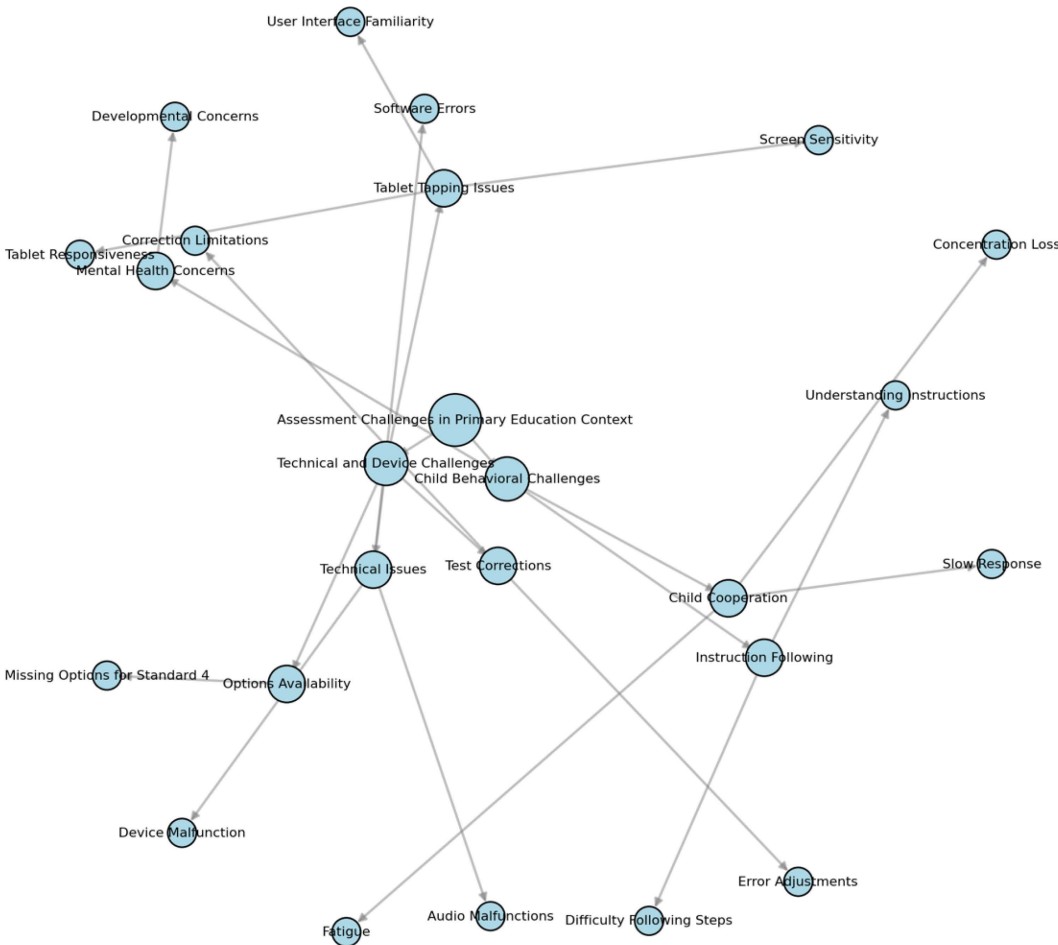

**Fig 4. Thematic Network Analysis of Recurring Assessment Challenges Themes. Nodes indicate specific assessor comments.** Vectors identify thematically similar comments.

time metrics to address this issue (see S1 Fig, S2 Fig), we recommend increasing the speed at which individual items are presented to mitigate this problem in the future. Fourth, this is our first time using the Short Memory task. Unlike other tasks that involved a fixed items length, the number of trials depended on student performance. As such, student scores are based on varying number of items, which introduces the potential for differential precision of measurement. We recommend modifying this task to include a fixed set of items.

Previous studies evaluating the impact of malaria control in school-aged children on cognition have primarily evaluated sustained attention, using a single task (e.g., code transmission test) which requires writing numbers, making it more useful on older age groups [25]. Given the heterogeneous results of these studies and an awareness of the contribution of EF to learning, a broader set of EF tasks, as used in our study, may be useful to more comprehensively evaluate the impact of malaria prevention and control on cognition and learning in primary school-aged children.

This study had two primary limitations. First, we worked in a single primary school in rural Malawi, which may limit generalizability of results. Nonetheless, this school setting was similar to schools across Malawi and other malaria endemic regions in that student teacher ratios are high, age varies greatly within grade levels due to late enrollments as well as grade retention, literacy rates among adults in the community are low, and few students in study had prior exposure to

smart phones or tablet computers. Second, we focused on the feasibility and initial promise (e.g., distributional character-istics; age differences) of a revised battery of EF tasks for use with primary school-aged students. Future studies should consider the construct and predictive validity of this task battery, which will require the administration of additional EF tasks and standardized assessments of learning, such as the Early Grade Reading and Math Assessment.

In conclusion, this study demonstrates the feasibility and utility of incorporating direct tablet-based assessments of EF skills as outcomes in studies that assess malaria interventions in school-aged students. Many previous studies have relied on single performance-based tasks or observer report to index neurodevelopmental constructs, and more information is needed to establish the validity and reliability of existing measures [26]. We see merit in administering a battery of EF tasks that can be combined into subdomain (e.g., inhibitory control, working memory) or overall composite scores, which improves both the breadth of measurement and the reliability of scores. Optimizing the measurement of EF is essential if these skills are to be included as outcomes in future studies that evaluate the impact of malaria on cognition and learning, as well as interventions studies striving to mitigate that impact.

## Supporting information

**S1 Fig. Bivariate Association Between Accuracy-Only and Accuracy + Reaction Time Scores for Inhibitory Control Subscale of the Hearts & Flowers Task.** Factor scores that combine item-level information on accuracy and reaction time (y-axis) exhibit improved variation relative to accuracy only (x-axis) scores.
(TIF)

**S2 Fig. Univariate Distribution of Accuracy + Reaction Time Score for Inhibitory Control Subscale of the Hearts & Flowers Task.** The univariate distribution of combined accuracy + reaction time factor score preserves individual differences in task performance, even among students who complete 100% of items correctly.
(TIF)

**S1 Data. Deidentified data that was used in this study.**
(CSV)

**S1 Table. Elaborated description of executive function tasks.**
(TIF)

## Acknowledgments

We appreciate the contributions of Gilbert Phiri, Tatuna Ngwira, Fortunate Banda, Daniel Mulenga, and Friday Nantongwe who conducted child assessments and provided insightful comments regarding assessment challenges.

## Author contributions

**Conceptualization:** Michael T Willoughby, Maclean Vokhiwa, Richard Reithinger, Lauren M. Cohee.

**Data curation:** Maclean Vokhiwa, Amanda C. Wylie.

**Formal analysis:** Michael T Willoughby, Amanda C. Wylie.

**Funding acquisition:** Richard Reithinger.

**Investigation:** Michael T Willoughby, Maclean Vokhiwa, Richard Reithinger, Lauren M. Cohee.

**Methodology:** Lauren M. Cohee.

**Project administration:** Maclean Vokhiwa, Richard Reithinger.

**Resources:** Richard Reithinger.

**Software:** Amanda C. Wylie.

**Supervision:** Michael T Willoughby, Maclean Vokhiwa, Lauren M. Cohee.

**Visualization:** Amanda C. Wylie.

**Writing – original draft:** Michael T Willoughby, Maclean Vokhiwa.

**Writing – review & editing:** Amanda C. Wylie, Richard Reithinger, Lauren M. Cohee.

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
