## [Decision Letter · Decision Letter 0]

PGPH-D-25-00271

Testing the feasibility and utility of an executive function battery for use with primary school-aged students in Malawi

Dear Dr. Willoughby,

Thank you for submitting your manuscript to PLOS Global Public Health. After careful consideration, we feel that it has merit but does not fully meet PLOS Global Public Health’s publication criteria as it currently stands. Therefore, we invite you to submit a revised version of the manuscript that addresses the points raised during the review process.

I agree with Reviewer #1 that there needs to be strong justification for the validity and accuracy of the instrument before addressing feasibility and utility. Please address this major concern along with the other issues identified by the reviewer.

We look forward to receiving your revised manuscript.

Kind regards,

Sanghyuk S Shin

Academic Editor

Journal Requirements:

1. Please include a complete copy of PLOS’ questionnaire on inclusivity in global research in your revised manuscript. Our policy for research in this area aims to improve transparency in the reporting of research performed outside of researchers’ own country or community. The policy applies to researchers who have travelled to a different country to conduct research, research with Indigenous populations or their lands, and research on cultural artefacts. The questionnaire can also be requested at the journal’s discretion for any other submissions, even if these conditions are not met. Please find more information on the policy and a link to download a blank copy of the questionnaire here: https://journals.plos.org/globalpublichealth/s/best-practices-in-research-reporting. Please upload a completed version of your questionnaire as Supporting Information when you resubmit your manuscript. 2. Your current Financial Disclosure states, “This study was conducted using internal funds from RTI International.”. However, your funding information on the submission form indicates that you did not receive funding. Please indicate by return email the full and correct funding information for your study and confirm the order in which funding contributions should appear. Please be sure to indicate whether the funders played any role in the study design, data collection and analysis, decision to publish, or preparation of the manuscript. 3. We do not publish any copyright or trademark symbols that usually accompany proprietary names, eg (R), (C), or TM  (e.g. next to drug or reagent names). Please remove all instances of trademark/copyright symbols throughout the text, including TM on page 5 and in abstract. 4. In the online submission form, you indicated that Access to study data may be made available upon reasonable request to the contact author and at the discretion of the ethics review committee. All PLOS journals now require all data underlying the findings described in their manuscript to be freely available to other researchers, either 1. In a public repository, 2. Within the manuscript itself, or 3. Uploaded as supplementary information. This policy applies to all data except where public deposition would breach compliance with the protocol approved by your research ethics board. If your data cannot be made publicly available for ethical or legal reasons (e.g., public availability would compromise patient privacy), please explain your reasons by return email and your exemption request will be escalated to the editor for approval. Your exemption request will be handled independently and will not hold up the peer review process, but will need to be resolved should your manuscript be accepted for publication. One of the Editorial team will then be in touch if there are any issues.

Additional Editor Comments (if provided):

Reviewers' comments:

Reviewer's Responses to Questions

**Comments to the Author**

1. Does this manuscript meet PLOS Global Public Health’s publication criteria ? Is the manuscript technically sound, and do the data support the conclusions? The manuscript must describe methodologically and ethically rigorous research with conclusions that are appropriately drawn based on the data presented.

Reviewer #1: Partly

2. Has the statistical analysis been performed appropriately and rigorously?

Reviewer #1: No

3. Have the authors made all data underlying the findings in their manuscript fully available (please refer to the Data Availability Statement at the start of the manuscript PDF file)?

Reviewer #1: No

4. Is the manuscript presented in an intelligible fashion and written in standard English?

Reviewer #1: Yes

5. Review Comments to the Author

Reviewer #1: In this paper by Willoughby et al, assessed the feasibility of a tablet-based tool to assess executive function in Malawian children from grades 1 to 4. This is an interesting question that aims to address challenges in assessing developmental outcomes in children that don’t require hours of testing by highly specialized test administrators. The ability to create a game-based platform that is engaging and provides reliable information on cognition and executive function would be transformative. In this paper, the authors demonstrate that it is easy to assess many children within a short period in school-based settings using this tool. I have some concerns about the construct validity of this tool, and the relatively weak correlation between the tasks meant to reflect the same sub-domain. Additional studies are needed to evaluate the correlation between these executive function tasks and reference tools to demonstrate that the construct validity is retained, particularly when modifications are made.

Specific comments

The authors note that the goal of the work is to assess the feasibility of implementing a tablet-based test to assess executive function. I feel like they have demonstrated feasibility. However, there is a bigger question of whether this test is a valid assessment of executive function and the domains assessed and whether they retain their construct validity in the translation and in this population unaccustomed to touch-screen technology. This is a bigger question that is not addressed or even mentioned and seems like a significant oversight. Just because we can administer the test and get results that demonstrate reasonable dispersion and children are able to complete the tasks doesn't mean they are accurately measuring the domains of interest.

The authors generated the executive function composite by taking the mean of the tasks completed. Six tasks were included that represented inhibitory control, cognitive flexibility, and working memory. However, not all tasks were completed by all children. To generate a balanced score reflecting these three domains, the mean per domain should be calculated and then averaged to develop an overall composite.

In the challenges of executive function assessment, the authors note student behavioral challenges, including poor concentration, fatigue, and developmental concerns. Ideally, these assessments should be administered in a quiet environment where children are well-rested and can concentrate. Details on test administration (e.g., time of day) and the ability to take breaks are not provided.

In the discussion, the authors recommend modifications to the battery of executive function tasks to drop certain games based on the experiences noted and scores generated. However, there is no discussion about the impact of modifications on the validity of the tests or the need to ensure they are measuring what they say they are before implementation of the tool.

6. PLOS authors have the option to publish the peer review history of their article (what does this mean? ). If published, this will include your full peer review and any attached files.

**Do you want your identity to be public for this peer review?** For information about this choice, including consent withdrawal, please see our Privacy Policy .

Reviewer #1: No

---

## [Decision Letter · Decision Letter 1]

Testing the feasibility and utility of an executive function battery for use with primary school-aged students in Malawi

PGPH-D-25-00271R1

Dear Dr. Willoughby,

We are pleased to inform you that your manuscript 'Testing the feasibility and utility of an executive function battery for use with primary school-aged students in Malawi' has been provisionally accepted for publication in PLOS Global Public Health.

Best regards,

Sanghyuk S Shin

Academic Editor

Reviewer Comments (if any, and for reference):

Reviewer's Responses to Questions

**Comments to the Author**

1. If the authors have adequately addressed your comments raised in a previous round of review and you feel that this manuscript is now acceptable for publication, you may indicate that here to bypass the “Comments to the Author” section, enter your conflict of interest statement in the “Confidential to Editor” section, and submit your "Accept" recommendation.

Reviewer #1: All comments have been addressed

2. Does this manuscript meet PLOS Global Public Health’s publication criteria ? Is the manuscript technically sound, and do the data support the conclusions? The manuscript must describe methodologically and ethically rigorous research with conclusions that are appropriately drawn based on the data presented.

Reviewer #1: Yes

3. Has the statistical analysis been performed appropriately and rigorously?

Reviewer #1: Yes

4. Have the authors made all data underlying the findings in their manuscript fully available (please refer to the Data Availability Statement at the start of the manuscript PDF file)?

Reviewer #1: Yes

5. Is the manuscript presented in an intelligible fashion and written in standard English?

Reviewer #1: Yes

6. Review Comments to the Author

Reviewer #1: Thank you for your responsiveness to my comments. I have no further comments.

7. PLOS authors have the option to publish the peer review history of their article (what does this mean? ). If published, this will include your full peer review and any attached files.

**Do you want your identity to be public for this peer review?** For information about this choice, including consent withdrawal, please see our Privacy Policy .

Reviewer #1: No
